## [Peer Review File · Journal of Cell Science]

The nuclear envelope protein TMEM209 is an integral component of the nuclear pore complex and interacts with Nup210

David Kohlhouse, Christiane Spillner, Violeta Alcalde Zapata, Christof Lenz, Henning Urlaub, Tobias Kohl, Stephan Elmar Lehnart, Larry Gerace and Ralph Heinz Kehlenbach
DOI: 10.1242/jcs.264534

Editor: Megan King

Review timeline

Original submission:	28 October 2025
Editorial decision:	3 December 2025
First revision received:	6 January 2026
Accepted:	8 January 2026

Original submission

First decision letter

MS ID#: jcs.264534

MS TITLE: The nuclear envelope protein TMEM209 is an integral component of the nuclear pore complex and interacts with Nup210

AUTHORS: Ralph Heinz Kehlenbach; David Kohlhouse; Christiane Spillner; Violeta Alcalde Zapata; Christof Lenz; Henning Urlaub; Tobias Kohl; Stephan Elmar Lehnart; Larry Gerace

ARTICLE TYPE: Research Article

Dear Ralph,

We have now reached a decision on the above manuscript.

As you will see, the reviewers are generally enthusiastic about the manuscript and have predominantly minor comments. It would be nice for you to address (or at least comment) on the question raised by Reviewer 1 about the effect of combined knock-down of TMEM209 and NUP210 on NPC assembly. I hope that you will be able to easily address the remaining comments because I would like to be able to accept your paper, which I expect I can assess without further comments from reviewers.

Reviewer 1

Advance summary and potential significance to field

The manuscript by Kohlhouse et al. identifies the nuclear envelope transmembrane protein TMEM209 as a structural component of the nuclear pore complex (NPC), closely associated with—if not directly interacting with—NUP210. To date, only three transmembrane proteins have been firmly associated with the NPC (NUP210, POM121, and NDC1). Notably, knockdown of any one of these three is non-lethal in cells and in vivo, suggesting the existence of additional membrane-integrated components with overlapping or redundant functions. The authors' findings are interesting, align with observations from other systems (including plants), and provide novel insight

into NPC composition in human cells. Overall, the study is appropriate for publication in JCS, but the following points would strengthen the manuscript:

Major comments

1. According to AI-based interaction predictions (<http://prodata.swmed.edu/humanPPI/>), GP210 is a high-confidence interaction partner for TMEM209 corroborating the author's conclusions. It is suggested to at least discuss this. While not required, it may be relatively easy to design mutations that interrupt the predicted interaction and add this to the IP experiments. Including it would make for a more definitive study in the opinion of this reviewer. Similarly, it would be informative to examine the localization of such mutant(s).
2. The authors show that loss of TMEM209 leads to a modest reduction in proliferation, whereas combined depletion of TMEM209 and NUP210 results in a striking proliferation defect. What remains unclear, however, is the effect on NPC assembly. Knockdown of NUP210 or TMEM209 alone does not appear to impair assembly (as suggested for TMEM209 in Figure 5C and S3), and the same holds for POM121. Even combined depletion of NUP210 and POM121 produces functional NPCs, whereas only NDC1 knockdown disrupts NPC assembly. The authors attribute this robustness to redundancy among TM-Nups. If so, what happens when TMEM209 and NUP210 are depleted together? And what happens if POM121 is additionally knocked down? These experiments could reveal whether these TM-Nups collectively form a vital, functionally essential module for NPC assembly—or not.
3. Figure 1A (optional): While the distributions seen from the IF localization of HA-TMEM209 and TMEM209-HA were similar, slight differences were noted regarding the localization patterns seen in Figure 1A. While these differences could just be due to heterogeneous expression patterns, they could alternatively be explained due to altered expression levels, or potentially, partial translation or cleavage of the N-terminal HA. It may be worthwhile to include a western blot (or even quantification through integrated cell intensity of current IF images) to show potential expression differences between HA-TMEM209 vs TMEM209-HA.

Minor comments

1. Figure 1B: It would be useful to include both the AlphaFold pLDDT prediction score of TMEM209, as well as the transmembrane-specific CCTOP result reliability, qValue, and evaluation result (e.g., "good") scores.

Reviewer 2

Advance summary and potential significance to field

The paper gives a comprehensive and clearly documented account of the localisation and close proximity of the nuclear envelope transmembrane protein TMEM209 to nuclear pore complex (NPC) proteins particularly the membrane protein Nup210. The analysis is based on MS-based and IP-westerns based co-enrichment as well as IF and STED imaging based co-localisation. Knowing the composition of human NPCs is important and for understanding their function. TMEM209 had not yet been firmly established as an NPC component, even though, as nicely discussed by the authors, also comparison to other organisms supports this claim.

The results are clearly described and the discussion makes an interesting read. The dissociation of Nup210 upon overexpression of TMEM209 is interesting.

--Manuscript Draft--

Comments for the author

Major comments [Please request additional experiments only if they are essential for supporting the conclusions; authors should be encouraged to highlight any claims that are preliminary or speculative, or to discuss any pitfalls or alternative interpretations in a 'Limitations' section]

I have no major comments

Minor comments

Please provide clarification about the number of repetitions for the IP experiments reported; I could not find the statistics related to the IPs.

Please provide clarification about the AB titers used in the PLA assays, and also here I missed a record of the number of repetitions.

please explain why in S2 the 90 degrees rotated control (nup153-nup210) is not 50-50 like it is for the main text figure 3F (nup153-TMEM209). please provide statistics to the claim "largely reduced overlap" in these figures.

It would be nicer if the alpha fold structure in figure 1 would be better connected to the different truncations used in figure 7.

First revision

Author response to reviewers' comments

Dear Megan,

Thank you very much for giving us the opportunity to submit a revised version. We have addressed all of the reviewers' comments, incorporated new data (novel Figures S3 and S7), an AlphaFold 3 model (Fig. S6), and revised several sections of the text to enhance clarity and readability. We also corrected a mistake in Figure 2C (lanes for TMEM209 and emerin had been mixed up) and include a few results in this letter (for reviewers only). Our point-by-point responses to the reviewers' comments are provided below in blue. Finally, we will upload the figures (unchanged) again at a higher resolution.

Reviewer 1:

The manuscript by Kohlhouse et al. identifies the nuclear envelope transmembrane protein TMEM209 as a structural component of the nuclear pore complex (NPC), closely associated with—if not directly interacting with—NUP210. To date, only three transmembrane proteins have been firmly associated with the NPC (NUP210, POM121, and NDC1). Notably, knockdown of any one of these three is non-lethal in cells and in vivo, suggesting the existence of additional membrane-integrated components with overlapping or redundant functions. The authors' findings are interesting, align with observations from other systems (including plants), and provide novel insight into NPC composition in human cells. Overall, the study is appropriate for publication in JCS, but the following points would strengthen the manuscript:

Major comments

1. According to AI-based interaction predictions (<http://prodata.swmed.edu/humanPPI/>), GP210 is a high-confidence interaction partner for TMEM209 corroborating the author's conclusions. It is suggested to at least discuss this.

Thank you very much for bringing this paper (which was published at the time of our first submission) to our attention. We now mention the predicted interactions in the discussion and cite the paper by Zhang et al.

While not required, it may be relatively easy to design mutations that interrupt the predicted interaction and add this to the IP experiments. Including it would make for a more definitive study in the opinion of this reviewer. Similarly, it would be informative to examine the localization of such mutant(s).

Indeed, such experiments are planned for a follow-up project. Concerning the suggested mutations: one of the two predictions (Zhang et al.; see below) suggests that mainly the TMDs of both proteins are involved in the interaction. We already address this in our IP-experiments (Figures 7B and D).

Interaction model for TMEM209 (in blue) and Nup210 (in green) according to Zhang et al. The blue helices correspond to the ones seen in Figure 1B. The interacting green helix corresponds to the single TMD of Nup210. Compare (novel) Figure S6, which shows the AlphaFold 3 model of the same protein pair.

Loaded structure: Q8TEM1_S3__Q96SK2_S0

The other prediction does not really take the topologies of Nup210 and TMEM209 into account (only a small proportion of Nup210 at its C-terminal end would be available for interaction with TMEM209, as the majority of the protein resides in the space between INM and ONM). Hence, some of the interactions suggested by the predictions are not really meaningful.

2. The authors show that loss of TMEM209 leads to a modest reduction in proliferation, whereas combined depletion of TMEM209 and NUP210 results in a striking proliferation defect. What remains unclear, however, is the effect on NPC assembly. Knockdown of NUP210 or TMEM209 alone does not appear to impair assembly (as suggested for TMEM209 in Figure 5C and S3), and the same holds for POM121. Even combined depletion of NUP210 and POM121 produces functional NPCs, whereas only NDC1 knockdown disrupts NPC assembly. The authors attribute this robustness to redundancy among TM-Nups. If so, what happens when TMEM209 and NUP210 are depleted together?

To further investigate potential effects on NPC assembly, we depleted Nup210 in HAP1 parental and TMEM209 knockout cells and performed indirect immunofluorescence to detect several nucleoporins and INM- proteins. As shown in the novel Supplementary Figure S7, the depletion of Nup210 in addition to the TMEM209 knockout does not affect the analyzed proteins (including POM121).

And what happens if POM121 is additionally knocked down? These experiments could reveal whether these TM-Nups collectively form a vital, functionally essential module for NPC assembly—or not.

We tried knocking down Nup210 *and* Pom121 in HAP1-knockout cells. For Pom121, a strong knock down could not be achieved (see figure below; quantification of three independent experiments after Western blotting). Likewise, a triple knockdown in HeLa cells using siRNAs was inefficient, thus preventing a definitive conclusion.

3. Figure 1A (optional): While the distributions seen from the IF localization of HA-TMEM209 and TMEM209-HA were similar, slight differences were noted regarding the localization patterns seen in Figure 1A. While these differences could just be due to heterogeneous expression patterns, they could alternatively be explained due to altered expression levels, or potentially, partial translation or cleavage of the N-terminal HA. It may be worthwhile to include a western blot (or even quantification through integrated cell intensity of current IF images) to show potential expression differences between HA-TMEM209 vs TMEM209-HA.

Indeed, in these transient transfection experiments, the expression levels vary from cell to cell and from experiment to experiment. The localization pattern of HA-TMEM209 and TMEM209-HA, however, was very similar (see additional cells below for both constructs). This is now mentioned in the legend to Figure 1.

In the meantime, we also generated a new construct coding for TMEM209-HA (based on the original one) with higher expression levels, allowing detection by Western-blotting. The protein had the expected size of 65 kDa and truncated versions were not detected (see below).

Minor comments

1. Figure 1B: It would be useful to include both the AlphaFold pLDDT prediction score of TMEM209, as well as the transmembrane-specific CCTOP result reliability, qValue, and evaluation result (e.g., "good") scores.

As suggested, we now include the additional information. Compare also Figure S6.

Reviewer 2:

The paper gives a comprehensive and clearly documented account of the localisation and close proximity of the nuclear envelope transmembrane protein TMEM209 to nuclear pore complex (NPC) proteins particularly the membrane protein Nup210. The analysis is based on MS-based and IP-westerns based co-enrichment as well as IF and STED imaging based co-localisation. Knowing the composition of human NPCs is important and for understanding their function. TMEM209 had not yet been firmly established as an NPC component, even though, as nicely discussed by the authors, also comparison to other organisms supports this claim.

The results are clearly described and the discussion makes an interesting read. The dissociation of Nup210 upon overexpression of TMEM209 is interesting.

I have no major comments

Minor comments

Please provide clarification about the number of repetitions for the IP experiments reported; I could not find the statistics related to the IPs.

We now provide this information in the legend to Figures 4 and 7. We are also adding new data (Figure S3): using digitonin as an alternative detergent for solubilization, we detected Nup205 as a protein that co-precipitates with endogenous TMEM209. Since Nup205 had been described as a TMEM209-interacting protein, we thought that this is worth mentioning.

Please provide clarification about the AB titers used in the PLA assays, and also here I missed a record of the number of repetitions.

We now provide this information in the legend to Figure S1 and in Table S4.

please explain why in S2 the 90 degrees rotated control (nup153-nup210) is not 50-50 like it is for the main text figure 3F (nup153-TMEM209). please provide statistics to the claim "largely reduced overlap" in these figures.

We try to better explain this rotation control. The absolute values in the two individual experiments are not informative. The important point is that the percentage of "colocalizing

signals” (yellow part of the pie) is reduced upon rotation. This change is probably more obvious (down to 40.1%) in the control (Fig. S2) because of the higher quality of the antibody signals (endogenous Nup210 in S2 versus overexpressed HA-TMEM209 in Fig. 3F).

It would be nicer if the alpha fold structure in figure 1 would be better connected to the different truncations used in figure 7.

We refer to the respective figures now in the legends (Figures 1B, 7A) and also in the text.

We would like to thank the reviewers for their constructive comments and hope that we were able to further improve the quality of the manuscript.

With kind regards and best wishes for 2026,

Ralph

Second decision letter

MS ID#: jcs.264534R1

MS Title: The nuclear envelope protein TMEM209 is an integral component of the nuclear pore complex and interacts with Nup210

Authors: Ralph Heinz Kehlenbach; David Kohlhause; Christiane Spillner; Violeta Alcalde Zapata; Christof Lenz; Henning Urlaub; Tobias Kohl; Stephan Elmar Lehnart; Larry Gerace

Article Type: Research Article

Dear Ralph,

Happy New Year and thank you again for sending this exciting work to JCS. I am happy to tell you that your manuscript has been accepted for publication in Journal of Cell Science, pending standard publication integrity checks.